# Cognitive Issues in Pediatric Multiple Sclerosis

**DOI:** 10.3390/brainsci11040442

**Published:** 2021-03-30

**Authors:** Emilio Portaccio, Ermelinda De Meo, Angelo Bellinvia, Maria Pia Amato

**Affiliations:** 1Section Neurosciences, Dipartimento di Neuroscienze, Psicologia, Area del Farmaco e Salute del Bambino, University of Florence, 50139 Florence, Italy; angelo.bellinvia@unifi.it (A.B.); mariapia.amato@unifi.it (M.P.A.); 2IRCCS Fondazione Don Carlo Gnocchi, 50143 Florence, Italy; 3Neuroimaging Research Unit, Institute of Experimental Neurology, Division of Neuroscience, Vita-Salute San Raffaele University, 20132 Milan, Italy; demeo.ermelinda@hsr.it

**Keywords:** pediatric multiple sclerosis, cognitive impairment, magnetic resonance imaging

## Abstract

Multiple sclerosis (MS) is one of the leading causes of disability in young adults. The onset of MS during developmental age makes pediatric patients particularly susceptible to cognitive impairment, resulting from both disease-related damage and failure of age-expected brain growth. Despite different test batteries and definitions, cognitive impairment has been consistently reported in approximately one-third of pediatric patients with MS. However, the lack of a uniform definition of cognitive impairment and the adoption of different test batteries have led to divergent results in terms of cognitive domains more frequently affected across the cohorts explored. This heterogeneity has hampered large international collaborative studies. Moreover, research aimed at the identification of risk factors (e.g., demographic, clinical, and radiological features) or protective factors (e.g., cognitive reserve, leisure activities) for cognitive decline is still scanty. Mood disorders, such as depression and anxiety, can be detected in these patients alongside cognitive decline or in isolation, and can negatively affect quality of life scores as well as academic performances. By using MRI, cognitive impairment was attributed to damage to specific brain compartments as well as to abnormal network activation patterns. However, multimodal MRI studies are still needed in order to assess the contribution of each MRI metric to cognitive impairment. Importantly, longitudinal studies have recently demonstrated failure of age-expected brain growth and of white matter (WM) and gray matter (GM) maturation plays a relevant role in determining cognitive dysfunction, in addition to MS-related direct damage. Whether these growth retardations might result in specific cognitive profiles according to the age at disease onset has not been studied, yet. A better characterization of cognitive profiles in pediatric MS patients, as well as the definition of neuroanatomical substrates of cognitive impairment and their longitudinal evolution are needed to develop efficient therapeutic strategies against cognitive impairment in this patient population.

## 1. Introduction

Multiple sclerosis (MS) is an autoimmune inflammatory and neurodegenerative disease of the central nervous system (CNS). Approximately 2–5% of patients with MS experience their first demyelinating attack prior to age 18 [1,2]. Pediatric MS patients are likely to have higher disease activity, as they experience two to three times as many relapses as their adult counterpart [3]. Despite more frequent relapses, the CNS of pediatric patients has proven to be more resilient to demyelinating attacks than adults [4], possibly due to the higher myelin repair capacity, which is likely to decrease with age [5]. This specific feature of the pediatric population may underlie the slower disability accrual, measured on the Expanded Disability Status Scale, observed in children and adolescents compared to adults with MS [1,6].

Nonetheless, the onset of MS during a period of life in which myelination is ongoing in the developing CNS and the child is acquiring cognitive competencies, can have dramatic consequences on neuropsychological functioning [7]. The potential impact of cognitive impairment in this population, affecting academic and social functioning and subsequent outcomes in adult life, has made the assessment of cognitive function an essential component of clinical evaluation in pediatric MS patients.

An understanding of the neuroanatomical basis of cognitive impairment in pediatric MS patients is a necessary prerequisite to develop efficient therapeutic strategies against cognitive decline. MRI has proven to be a powerful tool to investigate the neuroanatomical basis of cognitive impairment in adult MS patients. Through its application to pediatric patients, several potential substrates were also individuated in this population.

In this narrative Review, we aim to provide a comprehensive and up-to-date perspective of cognitive impairment in pediatric MS patients. We researched the main international databases for research articles published in international peer-reviewed journals and based on a minimum cohort of pediatric MS patients arbitrarily settled as >10 subjects. We selected the following keywords for the research: “multiple sclerosis” and “cognitive”, and “MRI, and “pediatric”, “paediatric” or “children”. Based on our research and expertise in the field, we are going to discuss the clinical correlates and the impact on everyday life of cognitive and mood-related difficulties in pediatric MS patients, as well as their long-term consequences. Moreover, we are going to analyze the role of MRI for understanding of the neuroanatomical basis of cognitive dysfunction in pediatric MS patients, summarizing the main advancements separately for each MRI metric and technique. Finally, we are going to discuss unmet needs and to provide some guidance for future research in the field.

## 2. Cognitive Impairment in Pediatric MS—Clinical Perspective

### 2.1. Assessment

Cognitive assessment of pediatric MS patients can be challenging due to the existence of different age- and sex-related developmental trajectories, which require comparison with different normative groups. Moreover, there is no specific consensus on which neuropsychological tests to use, and on what are the best tools for evaluating the functional impact of cognitive issues on subject’s social and academic functioning [7,8].

Brief cognitive batteries are increasingly used, nowadays, as they have proven to be reliable in detecting cognitive impairment in the pediatric population. Some of these batteries have been specifically developed for pediatric MS patients, such as the Brief Neuropsychological Battery for Children (BNBC) [9]. The BNBC is a 30-min evaluation that includes: the Symbol Digit Modalities Test (SDMT), to evaluate information processing speed (IPS); the Trail Making Test A/B, to evaluate attention and IPS; a vocabulary test from the Wechsler Intelligence Scale for Children (WISC), to evaluate the intelligence quotient (IQ); and the Selective Reminding Test (SRT), to evaluate verbal memory. A score under the 5th percentile relative to healthy controls (HC) showed a sensitivity and a specificity of 96% and 81%, respectively, in detecting cognitive impairment, compared to a gold standard more extensive cognitive assessment [9].

Another brief assessment tool, the MS Inventory for Cognition in Adolescents (MUSICADO) [10], was recently developed to evaluate cognition, fatigue, and loss of health-related quality of life in pediatric MS patients. The cognitive section of this scale was obtained by analyzing the performances of pediatric MS patients on eight different tests assessing verbal short- and long-term memory, language, attention, working memory, cognitive flexibility, IPS, all validated in the German language. The cognitive tests included in this scale are: the German version of the Word List Generation (WLG), (Regensburger Wortflussigkeitstest, RWT), exploring language and executive functions; the TMT-A, exploring attention and IPS; and the Digit Span Forward, exploring working memory. The combination of abnormal results of these tests, defined as a score of 1.5 standard deviations (SD) below the mean of HC scores, resulted in a specificity of 88.6% in detecting cognitive impairment compared to gold standard [10].

The SDMT is a reliable index to detect cognitive impairment in adult MS patients. Indeed, it is now recommended as a cognitive screening test in adult MS patients, alone or together with the Rey Auditory Verbal Learning Test (RAVLT) and the Brief Visuospatial Memory Test-Revised (BVMTR) [11]. Since it can be administered in children as young as 8 years, the SDMT is increasingly used as a screening test in pediatric MS patients. In a recent US-based study, pediatric patients with demyelinating diseases showed worse performance on the SMDT, compared to HC (Cohen’s d 1.30) [12]. In this same study, the SDMT yielded a 77% sensitivity and 81% specificity for detecting cognitive impairment in pediatric-onset MS patients who failed the SDMT and were evaluated with a more extensive neuropsychological battery within one year, and 100% specificity if the extensive battery was administered within two months [12]. Furthermore, in a recent longitudinal study including a large sample of pediatric MS patients (50 CIS and 333 MS), the SDMT alone allowed the detection of clinically-meaningful cognitive decline in 14.1% of the patients over 1.8 years [13].

Other studies adopted the Brief International Cognitive Assessment for MS (BICAMS) [12,14] and the Rao’s Brief Repeatable Battery (BRB) [15], commonly used in adults MS patients. The BICAMS explores attention and information processing speed (using the SDMT), visuospatial memory (using BVMTR), and verbal memory (using RAVLT). By adopting this battery cognitive impairment was observed in 26% of 69 pediatric MS patients enrolled in a US-based study [16].

More recently, normative values for the BRB have been published in an Italian adolescent population [17]. This battery explores the cognitive domains most frequently impaired in adult patients with MS, incorporating tests of verbal memory (SRT), complex attention and information processing speed (Paced Auditory Serial Addition Test and SDMT), verbal fluency (WLG) and visuospatial memory (10/36 Spatial Recall Test [SPART]). It has to be noted that the majority of available brief assessment tools do not explore executive functions, a domain frequently involved in pediatric MS patients (see “Prevalence and neuropsychological profile” section). Therefore, the administration of additional tests tapping these functions, such as measures belonging to the Delis-Kaplan Executive Function System [18] could be considered. In the last years, different computerized assessments for cognitive functioning have been developed and validated in adult patients with MS. Among these, the CogState Brief Battery (CBB) [19] has recently been applied to pediatric MS patients, showing a detection rate of cognitive impairment comparable to the BICAMS [16]. Furthermore, a computerized version of the SDMT was recently validated in a group of pediatric MS patients [20].

### 2.2. Prevalence and Neuropsychological Profile

Information on the prevalence and the profiles of cognitive impairment in pediatric MS patients is derived from several cohort studies, often including small samples and adopting different test batteries and definitions of cognitive impairment (Table 1). Despite this heterogeneity, cognitive impairment is consistently reported in approximately one-third of pediatric MS patients. Like adult-onset MS patients, information processing speed (IPS), attention, verbal and visuospatial memory and executive functions are most frequently affected [7,20]. However, pediatric MS patients may differ from their adult counterpart for a relevant involvement of language and general intelligence [21].

In an Italian multicenter study [21], cognitive impairment was detected in 19/63 patients (31%), with prominent involvement of verbal memory (53%), visuospatial memory (56%), complex attention and IPS (50%), and executive functions (41%). Cognitively impaired (CI) MS patients also experienced deficits in semantic (22%) and phonemic (17%) verbal fluencies and in two different verbal comprehension tasks (28% and 39%).

In a large sample from the US network of MS centers [23], cognitive impairment was observed among 35% of pediatric patients with relapsing-remitting MS and 18% of pediatric patients with clinically isolated syndrome (CIS). In this cohort, the cognitive functions more frequently involved were fine motor coordination (54%) and visuomotor integration (50%), followed by IPS (35%).

### 2.3. Evolution

Information about the long-term evolution of cognitive functioning in pediatric MS is still scanty [25]. Heterogeneous findings were reported, likely due to differences in patients’ clinical features, assessment tools and length of the follow-up period (Table 2). A relevant aspect is the definition of a meaningful change in pediatric cases. As a matter of fact, both reduced cognitive performance over time and loss of age-expected cognitive gains compared with healthy peers should be expected in the pediatric population. In a Canadian 15-month longitudinal study of 28 pediatric MS cases, improvements in cognitive functioning were reported on 18% of cognitive measures in the MS group, as compared with 86% of measures in HC [26]. Considering only significant declines on test scores, deterioration in functioning was observed in 7 of 28 patients (25%) as compared to only 1 of 26 HC (4.6%).

In an Italian cohort [30], 56 patients with pediatric MS were re-assessed after 2 years, and deterioration of cognitive performance was observed in the 75% of cases. The changes observed prominently affected verbal memory, complex attention, verbal fluency and receptive language.

Forty-eight subjects, belonging to this same cohort were re-assessed after 5 years [28], showing a deterioration of cognitive performance in 56% of the cases, as opposed to an improvement in 25% and stability in 19%. Functions most prone to deteriorate were visuospatial memory, verbal fluency, and expressive language.

On the other hand, in a US-based study [29] including 62 pediatric patients with MS and 5 with CIS, cognitive functioning remained stable in the majority of patients after a 1.6-year follow-up, while deterioration was detected in 13% and improvement in 20% of participants. In a large study from the US Network of Pediatric MS Centers assessing cognition by using the Symbol Digit Modalities Test (SDMT) only, performance declined in 14.1% of pediatric MS patients over 1.8-year follow-up [13].

Another relevant aspect to consider is the potential impact of MS onset during childhood on cognitive and functional attainments later in adulthood. An Italian cross-sectional multicenter study compared cognitive performance in adulthood between a group of 119 pediatric-onset MS and 712 adult-onset MS patients [31]. While at univariate analysis the proportion of cognitive impairment did not differ between the two groups (44.5% in pediatric MS vs. 48% in adult-onset MS), after adjustment for age, MS onset during childhood was associated with an increased risk for cognitive impairment in adulthood (odds ratio = 1.71). Likewise, in a population-based study performed in Sweden and including 300 pediatric-onset MS and 5407 adult-onset MS patients [32], those with a pediatric-onset demonstrated a more rapid decline over time in IPS performance and were more likely to experience cognitive impairment. These findings could be partially explained by failure of age-expected cognitive maturation due to disease-related changes in a critical period of life [32].

On the whole, while cognitive deterioration appears to be relatively frequent in pediatric MS, many patients may exhibit stable or improved cognitive functioning over time. In this perspective, according to recent recommendations on cognitive assessment in MS [11].

### 2.4. Clinical Correlates and Functional Impact

There is little information on clinical correlates of cognitive impairment in pediatric MS. In cross-sectional studies, a worse cognitive performance was mainly associated with younger age at onset [23,24], longer disease duration [13,24], higher disability levels [13,22,24] and higher number of relapses in the year preceding cognitive assessment [22].

The available evidence about potential predictors of cognitive decline over time is even more limited and sometimes conflicting in pediatric MS. Younger age at disease onset was associated with a higher likelihood of cognitive decline on measures of IPS [6,33,34] and working memory [34]. However, an older age at disease onset and male sex were also found to be associated with higher risk of cognitive decline assessed by using the SDMT [13].

In summary, there is a dearth of studies on predictors of cognitive decline over time in POMS patients, with conflicting and insufficient evidence about age at onset and sex as potential predicting factors.

A number of potential protective factors against cognitive decline have been suggested. In an Italian 5-year longitudinal study, higher IQ scores were associated with stable or improving cognitive performance at subsequent evaluations, particularly in cognitively-preserved subjects at the first assessment [35]. Positive effects of higher premorbid IQ in pediatric MS subjects appeared to be maintained even during adulthood [36,37]. These findings underscore the potential protective role of cognitive reserve against neuropsychological deterioration in pediatric MS. Indeed, cognitive reserve could be particularly efficient in children, who are deemed to have greater capacity to compensate from brain damage through neural plasticity. Therefore, the promotion of intellectual enrichment in pediatric patients should prevent the onset and/or the evolution of MS-related cognitive dysfunction.

The occurrence of cognitive impairment in pediatric MS patients can have a significant functional impact. In the US-based study [22], 35.1% of the pediatric MS patients required some type of aid or adaptation in their school curriculum. In the Italian multicentric study [28], at baseline evaluation, parents reported that for one-third of patients cognitive problems negatively affected school, daily and social activities. Longitudinal data of the same cohort over 5 years [29] demonstrated cognitive decline for most participants, associated with adverse consequences on school achievements and social life. In a population-based study in Denmark, while school performance in children with MS was not different as compared with that of children from the general population and children with other chronic diseases, a lower proportion of patients with pediatric-onset MS attended high school [38]. Finally, in a recent study comparing pediatric-onset and adult-onset Italian MS patients in adulthood, pediatric-onset MS patients exhibited a lower-than-expected educational level and they had a lower median premorbid IQ (101 vs. 106.5; *p* = 0.03) [36]. Lower educational level and IQ scores, together with higher disability negatively affected socio-professional attainment in terms of occupational complexity and unemployment rate [36]. Likewise, in a prospective register-based cohort study of 485 pediatric-onset MS patients, as compared with a population-based matched reference cohort (*n* = 4850), pediatric-onset MS was associated with less educational achievement, lower earnings, and greater use of disability benefits in adulthood [39].

### 2.5. Mood Disorders

Depression is present in up to 40% of adult MS patients [40], and pediatric patients with cognitive impairment may be at higher risk for mood disorders [27,41,42].

In general, parents are more prone to report mood disorders in their children as compared with patients’ self-report [43]. For instance, in a longitudinal study conducted over five years, parents reported behavioral problems in 39% of the included patients, and neuropsychological evaluation detected an affective disorder in 30% of unselected children [29]. In another study, according to parents’ reports, depressive symptoms were significantly more frequent in a group of children with demyelinating disorders as compared with HC (28.6% vs. 10.9%) [44]. Further studies confirmed the presence of a higher rate of depression in children with MS as opposed to both peers with other acquired demyelinating disorders (24% vs. 18%) and HC (22% vs. 11%) [23,45].

Mood disorders in pediatric-onset MS patients are not without consequences. Depression and anxiety can negatively affect quality of life scores along with fatigue, disability, and disease duration [41,46]. Moreover, children with MS and concurrent mood or anxiety disorders had a higher rate of cognitive dysfunction, compared with those with other psychiatric diagnoses [41,47].

In light of above evidence, systematic evaluation of mood disturbances, together with cognitive abilities, is of critical importance in order to provide prompt and appropriate interventions.

## 3. Cognitive Impairment in Pediatric MS—The MRI Perspective

### 3.1. MRI Role in Pediatric MS

MRI plays a pivotal role in the diagnosis of MS, in monitoring disease progression and treatment response, so much in children as it does in adults. In details, current goals of MRI in pediatric MS include: confirmation of a diagnosis of MS before a second clinical attack [48] in individuals with an acute demyelinating syndrome [49,50]; exclusion of alternative diagnoses [51]; and prediction of prognosis. In addition, serial MRI monitoring to qualitatively evaluate new lesion formation is useful for diagnosis, to inform on treatment decisions, and to monitor disease evolution. The formation of confluent lesions and atrophy are also increasingly being evaluated, qualitatively. However, conventional MRI measures only show weak associations with clinical features. Advanced MRI technique have greater means to provide insights on underlying pathological changes, which in turn will relate more directly to clinical and cognitive outcomes in this population. Table 3 summarizes results from the main studies aimed at identifying the pathophysiological mechanisms underlying cognitive impairment in pediatric MS patients, which we are going to discuss in the next paragraphs.

### 3.2. White Matter Lesion Features and Distribution

MRI scans of pediatric MS are typically characterized by multiple white matter (WM) lesions. In adolescents, the characteristics of these lesions are similar to those reported in adults: frequently located asymmetrically in the periventricular and juxtacortical WM regions, the corpus callosum and infratentorial regions, and characterized by oval or elliptical shapes [66]. To the opposite, in very young patients, T2-bright lesions more frequently have ill-defined borders or marked perilesional edema [67]. Furthermore, a significant number of these lesions vanishes on follow-up imaging, suggesting that demyelinating lesions may be of different nature in younger compared to older children. Direct comparisons of lesion distribution and volume between pediatric and adult-onset MS cohorts are rare in the literature [68]. Compared to adult patients, similar or higher volumes T2-hyperintense and T1-hypointense lesions have been reported in pediatric MS patients [68,69,70]. T2-lesion distribution in the supratentorial encephalon was similar in pediatric patients at their first demyelinating attack compared to adults with MS [70]. This observation implies that lesion accrual may not require a prolonged period of subclinical disease.

However, a preferential involvement of infratentorial regions by lesions was described in pediatric MS, with pontine lesions being particularly prominent in male patients [68,70]. These differences in lesion prevalence and location might reflect immunological differences, as well as differences in the stages of myelination. Myelination in the brainstem, including the pons, is retarded relative to the supratentorial WM, as it proceeds along a caudo-rostral gradient [71]. Moreover, the pons myelinates faster in males than in females [72], which seems to be consistent with the observation of preferential involvement of this structure in boys compared to girls. MS appears to target WM areas with more mature myelination in children.

Higher lesion volumes were associated with cognitive impairment in pediatric MS patients [52]. Moreover, anatomo-functional correlations were demonstrated. For instance, the salient involvement of linguistic abilities in the pediatric population [23] is consistent with a higher infratentorial lesion burden, affecting the afferent and efferent cerebellar pathways in children. The latter, in turn, was demonstrated to significantly affect performance in vocabulary tests [53].

### 3.3. Normal Appearing White Matter Damage

In addition to focal WM lesions, regions of normal appearing white matter (NAWM) often contain abnormalities in MS, including axonal spheroids and swellings, mild inflammation, microglial activation, gliosis, and increased expression of proteolytic enzymes [73]. These abnormalities go undetected with conventional MRI, partly explaining in part why only modest correlations between MRI-visible focal WM lesions and neurologic deficits were reported [74].

Diffusion tensor imaging (DTI) represents a powerful tool to assess NAWM microstructural integrity. By using this MRI technique, significant associations were found between NAWM abnormalities and cognitive functioning in pediatric MS patients. Till and colleagues [54] explored the relationship between academic functioning and WM integrity among children with MS compared with age and sex-matched HC. These Authors specifically analysed math performances, since they are strictly related to efficient IPS, working memory (e.g., carrying and borrowing digits), and visual-spatial processing (e.g., alignment of columns), all of which are commonly affected by MS. In this study, difficulties in written arithmetic ability were observed in 26% of patients and they were significantly associated with abnormalities in DTI metrics across all segments of the corpus callosum and in right frontal and parietal regions. A subsequent study [52] extended these findings, associating cognitive impairment in pediatric MS patients with DTI metrics abnormalities in posterior corpus callosum and cingulum as well as in bilateral parieto-occipital regions. Importantly, it should be reinforced that the abnormalities in NAWM might be attributed to either direct disease-related damage or failure of WM maturational processes, or a combination of both. As matter of fact, the onset of MS during childhood has proven to lead to failure of age-expected WM maturational processes [75]. So far, however, MRI studies have not been able to separate these aspects.

### 3.4. Gray Matter Lesions

By suppressing the signal from both WM and cerebrospinal fluid (CSF), double inversion recovery (DIR) sequences provide a good lesion contrast in the grey matter (GM), thus improving the detection of cortical lesions *in vivo* [76]. Compared to direct pathologic assessment, DIR sequences are able to detect only 10%–20% of cortical lesions in MS patients [77]. Nevertheless, MS patients have multiple cortical lesions, resulting in a relatively large prevalence of at least one cortical lesion on DIR sequences. A first study [78] investigating cortical lesions in 24 pediatric patients with relapsing-remitting MS in comparison to 10 adult relapsing-remitting MS patients found that less than 10% of pediatric patients had cortical lesions, in comparison to 66% of adults with MS [78]. These results were confirmed by two other studies (12% [56] in a group of 41, and 34% [79] in a group of 35 pediatric-onset MS patients). Despite the relevance of the cortex in cognitive functioning, only one [56] of these studies explored the role of cortical lesions in determining cognitive impairment in this population, without finding any association. Nevertheless, by 3T multiparametric techniques and 7T MRI have increased capacities to detect cortical lesions, finding them in 79% of 24 and 100% of 8 pediatric-onset MS patients, respectively [57,80]. Frontal lobe cortical lesion count was associated with reduced manual dexterity in pediatric-onset MS patients [57]. 

### 3.5. Gray Matter Damage

MS-related GM atrophy develops following a trajectory that starts from the deep GM nuclei and spreads, over time, to many cortical GM regions [81,82]. In this scenario, the thalamus is one of the first GM structures affected from the beginning of the disease, including patients with clinically isolated syndromes (CIS) [81,83] and pediatric MS [84,85,86]. Due to its central location between the WM and the CSF, it was demonstrated to be susceptible to heterogeneous pathological processes, including retrograde degeneration from WM lesions and CSF-mediated damage [87]. Furthermore, considering the role of the thalamus as a relay center provided with afferent and efferent connections with cortical and subcortical regions, it clearly is a key region for cognitive processes.

Till et al. [24] indicated that thalamic volume accounted for significant incremental variance in predicting global IQ, processing speed, and expressive vocabulary and was the most robust MRI predictor of cognitive impairment relative to other MRI metrics. Till et al. [58] also explored the association of executive dysfunction with structural MRI abnormalities in pediatric MS patients, finding significant correlations with thalamic volume.

Another determinant structure for cognition in pediatric MS was found to be the amygdala, whose lower volume was associated with a lower level of competency in functional communication skills [59]. The analysis of lateralized amygdala volumes revealed that the left amygdala was associated with both functional communication and social skills, consistent with literature suggesting the left amygdala has strong connections with emotional and language domains [88]. The volume of the amygdala also appeared to be associated with visual memory, thus suggesting a lateralized role of the right amygdala in memory for visual information.

Recently, Fuentes et al. [60] applied quantitative brain volumetric measures to better understand the neural correlates of learning and memory functioning in children and adolescents with MS. The Authors found significant associations of word-list learning with whole brain volume and hippocampal volume, whereas visual recognition memory correlated with thalamic volume.

The hippocampus is known to play a major role in cognitive processes. However, only one study [61] investigated the role of hippocampal damage in determining cognitive impairment in pediatric MS. In details, compared to cognitively preserved, cognitively impaired pediatric MS patients experienced atrophy of the right hippocampus at the level of the subiculum and the dentate gyrus. Moreover, significant correlations were found between performance at tests of language expression and comprehension with atrophy of subicular region of the right hippocampal head, the cornu ammonis 1 region of the right hippocampal tail and the dentate gyrus of the left hippocampal body. Furthermore, better performances on the TMT and the phrase comprehension test were associated with an increased volume of the dentate gyrus, bilaterally, suggesting that hypertrophy of this region might confer protection against the onset of cognitive deficits.

Moving to the cortex, atrophy of the precuneus was related with reduced cognitive performance in pediatric MS patients [52]. Indeed, this region is involved in a wide spectrum of highly integrated tasks, including visuospatial imagery, episodic memory, and self-processing operations [89].

Finally, the cerebellum represents a strategic node in various segregated networks (motor, coordination, cognitive–behavioral loops), showing multiple connections to and from different cortical areas of the forebrain, the thalamus, and the spinal cord. Within this framework, it is not surprising that cerebellar posterior lobe volume reduction can adversely impact cognitive function and especially information processing speed and vocabulary abilities in pediatric MS patients [53], which is consistent with the role of the posterolateral hemispheres of the cerebellum in cognitive processing [90].

### 3.6. Functional MRI

The poor correlation between structural damage and clinical phenotype might be explained by interindividual differences in brain capacity to respond to damage, in terms of both: recovery from tissue damage (remyelination) [91] and cortical plasticity [92]. Brain plasticity, and the derived functional reorganization, relies on molecular and cellular mechanisms, including increased axonal expression of sodium channels, synaptic changes, increased recruitment of parallel existing pathways or “latent” connections, and reorganization of distant sites [93]. These molecular and cellular alterations induce changes in systems-level functional responses, which are the proximal effectors of perception, action and cognition. The application of functional MRI (fMRI), based on changes in the blood-oxygenation level dependent (BOLD) signal, provides an indirect measure of neural activity, thus representing a powerful tool to measure brain plasticity in vivo. Two different paradigms of fMRI exist: task-dependent fMRI, requiring stimulus presentation; and resting-state (RS) fMRI, allowing to explore brain networks without performing any specific tasks.

Considering the difficulty to perform task-dependent studies in the pediatric population, only few studies adopting this paradigm have been conducted to date. Starting from the consideration that abnormalities in sustained attention are frequently associated with behavioral, learning, emotional, and cognitive difficulties in adolescence, and that attention is one of the most frequent areas of cognitive impairment in pediatric MS patients, De Meo et al. [62], investigated the relationship between cognitive impairment and sustained attention system recruitment abnormalities in these patients.

In details, sustained attention system activity was studied with fMRI during the Conners Continuous Performance Test (CCPT), and the structural integrity of the connections between brain regions relevant to the task was measured by using DTI. Paralleling the findings of volumetric analysis [52], the central role of the precuneus was confirmed, as the main area showing decreased activation during CCPT in MS patients compared to HC. However, global cognitive impairment in MS patients was attributed to an inefficient regulation of the functional interaction between different areas of the sustained attention system, due to abnormal WM integrity.

Another study conducted in cognitively preserved pediatric-onset MS aimed to assess the patterns of fMRI activity during the Alphaspan task [63], a working memory paradigm with two levels of executive control demand. In this study, cognitively preserved youth and young adults with pediatric-onset MS demonstrate greater activation than HC in regions implicated in executive control during a working memory task. These results support the hypothesis of increased brain activity as compensatory mechanism to maintain an adequate cognitive performance.

Contrary to task-dependent fMRI, multiple RS fMRI studies were performed. The analysis of RS functional connectivity (FC) within brain networks disclosed that functional abnormalities of precuneus paralleled abnormalities detected by structural MRI in pediatric patients with cognitive impairment [52,64]. Moreover, a distributed pattern of RS FC abnormalities within large-scale neuronal networks occurs in pediatric MS patients and contributes to their cognitive status. Focusing on the best studied RS cognitive network, the default mode network (DMN), increased RS FC in the anterior cingulate cortex was described in cognitively preserved, in addition to reduced RS FC in the precuneus in cognitively impaired pediatric MS patients [52]. Interestingly, RS FC abnormalities different between pediatric and adult MS patients with cognitive impairment. In adults, a consistent reduced RS FC of the anterior regions of the DMN [94,95] and an enhanced RS FC of the posterior ones have been described, contrary to children/adolescents [95,96]. It could be speculated that maturation effects might influence a different functional reorganization in adult vs. pediatric patients with MS, in response to WM damage. Indeed, long-range connections between the posterior cingulate cortex and the anterior prefrontal cortex have been shown to mature during late childhood/adolescence (being immature in 7-year-old children) [65,97], and to be associated with the development of cognitive abilities [71].

Similar to structural imaging, the cerebellum is another region whose RS FC plays a major role in determining cognitive functioning in pediatric MS patients. Indeed, compared to both HC and cognitively preserved pediatric MS patients, cognitively impaired patients showed a widespread reduction of RS FC, not only between the dentate nucleus and the basal ganglia, but also between the dentate nucleus and bilateral regions located in the parietal, frontal and temporal lobes [65].

## 4. Conclusions and Future Directions

Across differing test batteries and definitions of cognitive impairment, cognitive impairment is consistently reported in approximately one-third of pediatric patients with MS. It can have important negative consequences on everyday functioning and school performance as well as social and professional attainment in adulthood.

An early and systematic neuropsychological assessment is recommended in pediatric-onset MS for appropriate counselling and management. A prompt identification of subtle cognitive changes should promote the introduction of effective pharmacological and rehabilitative interventions, taking advantage of brain plasticity and compensatory abilities that are deemed to be more pronounced in this age range. In this regard, the assessment and potentiation of cognitive reserve through intellectual enrichment appears to be particularly relevant and promising. On the other hand, as for adult MS patients, cognitive measures should be included among clinical outcomes in trials evaluating the response to treatments.

Moreover, further studies focusing on relationships between cognitive functioning and brain MRI abnormalities are needed, especially with multimodal and longitudinal designs, which are bound to shed light on underlying pathophysiology. Despite several substrates of cognitive dysfunction have already been identified by MRI, further work is necessary from a translational point of view in order to establish a prognostic and monitoring value of MRI metrics in the field of MS-related cognitive impairment.

Longitudinal studies, including neuropsychological and multimodal MRI assessment, are mandatory in the pediatric MS population in order to unravel the complex interplay between brain development, MS-related brain damage and subsequent adaptive and maladaptive phenomena. A better characterization of cognitive impairment and of the underlying pathophysiological mechanisms might help individuate targets for novel pharmacological and rehabilitative treatment approaches, bringing hope for the future of this population.

## Figures and Tables

**Table 1 brainsci-11-00442-t001:** Main cross-sectional studies * on cognitive impairment in pediatric multiple sclerosis patients.

Study	Pediatric MS/HC Sample Size	Prevalence of Cognitive Impairment	Profile of Cognitive Impairment
McAllister et al., 2005 [22]	37 Pediatric MS patients/-	35%	Complex attention
Amato et al., 2008 [23]	63 Pediatric MS patients/57 HC	31%	Verbal and visual memory, complexattention, executive functions, language, IQ
Till et al., 2011 [24]	35 Pediatric MS patients/33 HC	29%	Attention and processing speed, visuomotor integration
Julian et al., 2013 [25]	187 Pediatric MS patients/-	35%	Fine motor speed, visuomotor integration, information processing speed
Wallach et al., 2020 [13]	616 Pediatric-onset MS patients/-	22%	Information processing speed §

* Full research articles published in international peer-reviewed journals, conducted on samples >10 subjects. § Single test assessment on the Symbol Digit Modalities Test. Abbreviations: MS = multiple sclerosis, HC = healthy controls.

**Table 2 brainsci-11-00442-t002:** Main longitudinal studies * on cognitive impairment in pediatric multiple sclerosis patients.

Study	Pediatric MS/HC Sample Size	FU Duration	Rate of Cognitive Deterioration
McAllister et al., 2007 [27]	12 Pediatric MS patients/-	3 years	42%
Amato et al., 2010 and 2014 [28,29]	48 Pediatric MS patients/-	5 years	56%
Till et al., 2013 [30]	28 Pediatric MS patients/26 HC	15 months	25%
Charvet et al., 2014 [31]	63 Pediatric MS patients/-	1.6 years	13%
Wallach et al., 2020 [13] §	383 Pediatric MS patients/-	1.8 years	14%

* Full research articles published in international peer-reviewed journals, conducted on samples > 10 subjects. § Single test assessment on the Symbol Digit Modalities Test. Abbreviations: MS = multiple sclerosis, HC = healthy controls.

**Table 3 brainsci-11-00442-t003:** Main MRI studies * aimed at identifying pathophysiological substrates of cognitive impairment in pediatric multiple sclerosis.

Brain Compartment	Technique	Study	Number of Participants with Pediatric MS	Analysis	Neuropsychological Assessment	Findings
WM	T2-PD and T1 lesion imaging	Rocca et al., 2014 [52]	35 pediatric MS patients (19 CP, 16 CI)	Lesion distribution on LPMs	Brief Neuropsychological Battery for Children [9]	CI MS patients had an increased probability of harboring lesions in the right thalamus, middle and posterior corpus callosum, and bilateral parieto-occipital regions
		Weier et al., 2015 [53]	28 pediatric MS patients	Lesion volumes	WASI, TMT– Part B; SDMT-oral version; Beery Visual Motor Integration	Infratentorial lesion volume affected performance in language and information processing speed.
NAWM	DTI	Till et al., 2011 [54]	31 pediatric MS patients	FA estimation in selected region of interest	WASI, SMDT-oral version and Woodcock–Johnson III Tests of Achievement [55]	Lower FA of in the corpus callosum correlated with math ability.
		Rocca et al., 2014 [52]	35 pediatric MS patients (19 CP, 16 CI)	Tract-based spatial statistic	Brief Neuropsychological Battery for Children [9]	Compared to CP, CI MS patients had decreased FA and increased RD of the posterior corpus callosum and cingulum as well as decreased FA of the bilateral parieto-occipital WM.
GM	DIR sequences	Rocca et al., 2015 [56]	41 pediatric MS patients (28 CP, 13 CI)	Cortical lesions identification	Brief Neuropsychological Battery for Children [9]	The number and volume of cortical lesions did not differ between CP and CI MS patients
	Multi-contrast 3T MRI	Maranzano et al., 2019 [57]	24 pediatric-onset MS patients	Cortical lesions identification	Matrix Reasoning and Vocabulary from the WASI, SDMT oral version, Decision Speed and Auditory Working Memory [55], TMT-Part A and B, and the RAVLT	Frontal lobe cortical lesions count was associated with reduced manual dexterity.
NAGM	3D T1 imaging	Till et al., 2011 [24]	34 pediatric MS patients (24 CP, 10 CI)	Atrophy measurements	WASI, SMDT oral version and Woodcock–Johnson III Tests of Achievement [55], TMT-Part A and B, CCPT, WSR from the Test of Memory and Learning—2nd edition, Beery–Buktenica Developmental Test of Visual Motor Integration—5th edition, phonemic Verbal Fluency subtest from the D-KEFS and WCST.	Thalamic volume accounted for significant incremental variance in predicting global IQ, processing speed, and expressive vocabulary and was the most robust MRI predictor of cognition.
	3D T1 imaging	Till et al., 2012 [58]	34 pediatric MS patients	Atrophy measurements	CCPT, Color-Word Interference Test from the D-KEFS, SMDT, Verbal Fluency test from the D-KEFS (Delis et al., 2001), WCST, TMT Part A and B.	Lower frontal lobe and thalamic volume correlated with poor performance on the TMT-B and Verbal Fluency
	3D T1 imaging	Green et al., 2018 [59]	32 pediatric MS patients	Atrophy measurements	WASI, TOMAL-2, MFS-D, WSR-D, FM and AVM.	Poorer memory was associated with reduced functional communication skills and reduced amygdala volume. Right amygdala volume was positively associated with visual memory; left amygdala volume was a strong predictor of parent-reported social skills.
	3D T1 imaging	Fuentes et al., 2012 [60]	32 pediatric MS patients	Atrophy measurements	WASI, TOMAL-2, MFS-D, WSR-D, FM and AVM.	Word-list learning correlated with whole brain volume and hippocampal volume, whereas visual recognition memory correlated with thalamic.
	3D T1 imaging	Rocca et al., 2016 [61]	53 pediatric MS patients (41 CP, 12 CI)	Radial mapping analysis of hippocampus	Brief Neuropsychological Battery for Children [9]	Compared to CP, CI MS patients had atrophy of the subiculum and dentate gyrus subfields of the right hippocampus.
	3D T1 imaging	Rocca et al., 2014 [52]	35 pediatric MS patients (19 CP, 16 CI)	Voxel based morphometry	Brief Neuropsychological Battery for Children [9]	Compared to CP, CI MS patients had atrophy of the right precuneus and left middle temporal gyrus. Compared to healthy controls and CP, CI MS patients had atrophy of the R precuneus.
	3D T1 imaging	Weier et al., 2015 [53]	28 pediatric MS patients	Atrophy measurements	WASI, TMT– Part B; SDMT-oral version; Beery Visual Motor Integration	Cerebellar posterior lobe volume influenced information processing and language performance.
	Task-based fMRI	De Meo et al., 2017 [62]	57 pediatric MS patients (44 CP, 13 CI)	Activation pattern analysis during CCPT	Brief Neuropsychological Battery for Children [9], CCPT	During CCPT, with increasing task demand, compared to CP, CI MS patients had decreased recruitment of several areas located mainly in parietal and occipital lobes and cerebellum and increased deactivation of the anterior cingulate cortex, combined with more severe structural damage of WM tracts connecting these regions.
	Task-based fMRI	Barlow-Krelina et al., 2019 [63]	20 CP pediatric-onset MS patients	Activation pattern analysis during Alphaspan task	WASI, RAVLT, Decision Speed and Auditory Working Memory, SDMT oral version, TMT.	Compared to healthy controls, CP MS patients experienced enhanced activation in the right middle frontal, left paracingulate, right supramarginal, and left superior parietal gyri during the low executive demand condition, over and above differences in response time. CP MS patients also demonstrated heightened activation in the right supramarginal gyrus in the high executive demand condition.
	Resting state fMRI	Rocca et al., 2014 [52]	35 pediatric MS patients (19 CP, 16 CI)	DMN functional connectivity	Brief Neuropsychological Battery for Children [9]	CI MS patients vs. both healthy controls and CP patients had decreased RS-FC of the right precuneus. Compared to both healthy controls and CI, CP MS patients experienced an increased RS-FC of the right anterior cingulate cortex.
	Resting state fMRI	Rocca et al., 2014 [64]	44 pediatric MS patients (25 CP, 19 CI)	Intra-network and inter-network FC	Brief Neuropsychological Battery for Children [9]	CI MS patients had decreased RS FC of the right precuneus of the left WMN, increased FC between the sensorimotor network and the DMN, and between the left WMN and the attention network as well as a decreased FC between left WMN and the DMN.
	Resting state fMRI	Cirillo et al., 2015 [65]	48 pediatric MS patients (35 CP, 8 CI)	RSFC using cerebellar dentate nucleus as seed region	Brief Neuropsychological Battery for Children [9]	CI MS patients showed a widespread reduction of RSFC of the dentate nucleus with basal ganglia and bilateral regions located in the parietal, frontal and temporal lobes.

* Full research articles published in international peer-reviewed journals, conducted on samples >10 subject. Abbreviations: WM = white matter; GM = gray matter; PD = proton density; CP = cognitively preserved; CI = cognitively impaired; MS = multiple sclerosis; LPMs = lesion probability maps; WASI = Wechsler Abbreviated Scale of Intelligence; SDMT = symbol digit modalities test; TMT = trail making test; FA = fractional anisotropy; RD = radial diffusivity; IQ = intelligence quotient; RAVLT = Rey Auditory Verbal Learning Test; CCPT = Conners Continuous Performance Test; WSR = Word Selective Reminding; WSR-D = Word Selective Reminding Delayed; WCST = Wisconsin Card Sorting Test; D-KEFS = Delis-Kaplan Executive Function System; TOMAL-2 = Test of Memory and Learning Second Edition, MFS-D = Memory for Stories Delayed; FM = Facial Memory; AVM = Abstract Visual Memory.

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
