# Peer review of "Cognitive Issues in Pediatric Multiple Sclerosis"

_brainsci, 2021, doi:10.3390/brainsci11040442_

Round 1
Reviewer 1 Report
The paper reviewed recent findings in cognitive issues related to pediatric multiple sclerosis (MS) patients focusing on the clinical perspective e.g. clinical assessment of cognitive impairment, clinical correlates and functional impact, and the MRI perspective.
The paper is well written and very informative for both clinicians and researchers in the MS field.
Table 3 is not easy to read the findings, should be revised.
Many typos should be corrected before publishing. A few examples, font size of first paragraph of the abstract is different to that of the 2nd paragraph. There should be a space before citing refs in text, e.g. line 41, there should be a space between “adults” and “[4]”.
Author Response
The paper reviewed recent findings in cognitive issues related to pediatric multiple sclerosis (MS) patients focusing on the clinical perspective e.g. clinical assessment of cognitive impairment, clinical correlates and functional impact, and the MRI perspective.
The paper is well written and very informative for both clinicians and researchers in the MS field.
A: we thank the reviewer for her/his appreciation of our work.
- Table 3 is not easy to read the findings, should be revised.
A: Table 3 has been revised according to the reviewer’s suggestion.
- Many typos should be corrected before publishing. A few examples, font size of first paragraph of the abstract is different to that of the 2nd There should be a space before citing refs in text, e.g. line 41, there should be a space between “adults” and “[4]”.
A: The manuscript was thoroughly revised, and typos corrected.
Reviewer 2 Report
Congratulations for your excellent work. I am sure it will be a valuable contribution in the field.
I have only two comments, which might be considered as recommendation for minor changes :
- A sentence, or at least a phrase for “mood disorders” could be included in the abstract, in agreement with the paragraph in the text dedicated for them.
- Reading about predictors of cognitive impairment in this population, one of them seems to be “younger age at onset”. It would be interesting if you clarify in the text if the literature you went through gives sufficient evidence to support this.
Author Response
Congratulations for your excellent work. I am sure it will be a valuable contribution in the field.
A: we thank the reviewer for her/his appreciation of our work.
I have only two comments, which might be considered as recommendation for minor changes:
- A sentence, or at least a phrase for “mood disorders” could be included in the abstract, in agreement with the paragraph in the text dedicated for them.
A: a brief sentence on “mood disorders” has been added in the abstract, lines 23-25.
- Reading about predictors of cognitive impairment in this population, one of them seems to be “younger age at onset”. It would be interesting if you clarify in the text if the literature you went through gives sufficient evidence to support this.
A: The paragraph about predictors of cognitive decline has been modified according to the reviewer’s suggestion (lines 219-221)